Facial expression morphing: enhancing visual fidelity and preserving facial details in CycleGAN-based expression synthesis

Sub-r-pa Chayanon
Chen Rung-Ching crching@cyut.edu.tw
Fan Ming-Zhong
Department of Information Management, Chaoyang University of Technology , Taichung , Taiwan
Yang Jiachen
Electronic publication date: 2024 Oct 25
Publication date: 2024
Volume: 10
Electronic Location ID: e2438
Received 2024 Jun 13; Accepted 2024 Sep 28
Copyright: ©2024 Sub-r-pa et al.
Copyright year: 2024
Copyright holder: Sub-r-pa et al.
License: This is an open access article distributed under the terms of the Creative Commons Attribution License, which permits unrestricted use, distribution, reproduction and adaptation in any medium and for any purpose provided that it is properly attributed. For attribution, the original author(s), title, publication source (PeerJ Computer Science) and either DOI or URL of the article must be cited.
License URL: https://creativecommons.org/licenses/by/4.0/

Keywords: Facial expression synthesis, GANs, Image-to-image, CycleGAN, Image processing, Image translation

Funding: NSTC, Taiwan Project No. NSTC-112-2221-E-324-003-MY3, NSTC-112-2221-E-324-011-MY2 This article is supported by the NSTC, Taiwan Project No. NSTC-112-2221-E-324-003-MY3 and NSTC-112-2221-E-324-011-MY2. The funders had no role in study design, data collection and analysis, decision to publish, or preparation of the manuscript.

==============================
Recent advancements in facial expression synthesis using deep learning, particularly with Cycle-Consistent Adversarial Networks (CycleGAN), have led to impressive results. However, a critical challenge persists: the generated expressions often lack the sharpness and fine details of the original face, such as freckles, moles, or birthmarks. To address this issue, we introduce the Facial Expression Morphing (FEM) algorithm, a novel post-processing method designed to enhance the visual fidelity of CycleGAN-based outputs. The FEM method blends the input image with the generated expression, prioritizing the preservation of crucial facial details. We experimented with our method on the Radboud Faces Database (RafD) and evaluated employing the Fréchet Inception Distance (FID) standard benchmark for image-to-image translation and introducing a new metric, FSD (Facial Similarity Distance), to specifically measure the similarity between translated and real images. Our comprehensive analysis of CycleGAN, UNet Vision Transformer cycle-consistent GAN versions 1 (UVCGANv1) and 2 (UVCGANv2) reveals a substantial enhancement in image clarity and preservation of intricate details. The average FID score of 31.92 achieved by our models represents a remarkable 50% reduction compared to the previous state-of-the-art model’s score of 63.82, showcasing the significant advancements made in this domain. This substantial enhancement in image quality is further supported by our proposed FSD metric, which shows a closer resemblance between FEM-processed images and the original faces.

Introduction

Facial editing (Botezatu et al., 2022) is now commonplace in social media, smartphone, and camera applications. It leverages an image processing algorithm that allows users to apply different effects to their images. For enhanced realism, many applications utilize technology such as generative adversarial networks (GANs) (Goodfellow et al., 2020; Karras et al., 2017), which utilize convolutional neural networks (CNN).

GANs are a machine learning framework comprising two neural networks, the generator, and the discriminator, engaged in a continuous feedback loop. The generator generates realistic data samples, such as images or text, attempting to deceive the discriminator. The discriminator, in turn, evaluates whether the data is genuine or fabricated by the generator. This adversarial training loop pushes both networks to improve, progressively enabling the generator to produce remarkably authentic data indistinguishable from real-world examples. GANs have revolutionized fields like image generation, art, and data augmentation, opening a new frontier of creative possibilities and practical applications.

Facial expression synthesis, a burgeoning field within computer graphics and machine learning, has experienced significant advancements due to the advent of GANs. This powerful framework, comprising a generator that crafts realistic facial images with specific expressions and a discriminator that distinguishes between genuine and synthetic faces, has proven effective in producing convincing facial expressions. The implications of this technology are promising to revolutionize various industries. From enhancing the realism of digital characters in video games and animations to potential applications in mental health therapies and human–computer interaction (Guo et al., 2020; Lian et al., 2020; Sun & Lv, 2019; Lou et al., 2019; Wang et al., 2022; Zhang et al., 2021), facial expression synthesis opens up new avenues for creative expression and communication. However, challenges such as maintaining identity consistency and avoiding the uncanny valley effect remain active research areas. As this field continues to evolve, facial expression synthesis is poised to reshape how we interact with digital faces and potentially even our understanding of human emotions.

The Latent Diffusion Model (LDM) (Rombach et al., 2022) is a state-of-the-art image generation technology, often producing better results than its predecessor, GANs. However, both LDM and GANs have difficulty with a common issue when creating facial expressions: they are good at making general faces but find it hard to accurately recreate the wide range of emotions and small details that make each expression unique. This challenge highlights the need for more research in this area, as being able to create facial expressions accurately

Facial expression synthesis research frequently involves image-to-image (I2I) translation (Isola et al., 2017), which leverages deep neural networks to modify images. These I2I models are designed to learn the complex task of transforming an image from one domain to another. This technology has demonstrated remarkable success in various applications, including transforming black-and-white images into vibrant color photographs and altering daytime scenes into evocative nighttime settings.

I2I translation typically requires paired input and output image datasets in the training process, which can be challenging to acquire in certain scenarios. For instance, image segmentation tasks require manual labeling or capturing the same scene under varying conditions. In the realm of facial expression synthesis, this challenge is compounded by the need for paired images of the same individual displaying different emotions, meticulously aligned and captured under consistent lighting. The unpaired I2I translation approach (Zhu et al., 2017; Torbunov et al., 2023b; Torbunov et al., 2023a; Zhao et al., 2022) offers a more flexible solution, proving particularly advantageous in this domain where achieving perfect alignment in paired data can be difficult due to the inherent dynamism of facial expressions.

A recent study (Sub-R-Pa & Chen, 2024) revealed that utilizing unpaired I2I translation with Cycle-Consistent Adversarial Networks (CycleGAN) (Zhu et al., 2017) and UVCGANv1 (Torbunov et al., 2023b) effectively transforms neutral and contemptuous facial expressions into happy ones. However, a notable drawback was observed: the resulting images often lacked crucial details from the original facial image Fig. 1. To address this issue, we propose a novel post-processing method called Facial Expression Morph (FEM), which merges the original facial image with the expression-transformed image while preserving essential features from both. This innovative approach aims to enhance the quality and realism of facial expression synthesis, ensuring that the resulting images retain the individuality and nuances of the original subject.

Figure 1 Using UVCGANv1 to transform “natural” to “happy”, (Sub-R-Pa & Chen, 2024) (A) full face, (B) detail in the top right, and (C) detail on the bottom left, with original quality.

(A∗), (B∗) and (C∗) Results from UCVGANv1 from the same portion, where quality is subpar due to unrealistic hair texture, eye rendering, and blurred jawline. Image source credit: Radboud Faces Database.

Advances in facial expression synthesis using deep learning, particularly with CycleGAN, have led to impressive results in generating realistic facial expressions. However, a critical challenge persists. While state-of-the-art models can effectively translate expressions, they often compromise the original facial identity. To address this issue, we hypothesize that by selectively combining the original image with only the expression area from the generated image, we can restore the facial identity while preserving the translated expression.

Our proposed FEM method is a novel approach that, following our hypothesis, combines image processing techniques with insights from psychology to improve facial expression synthesis. By selectively merging the original image with an expression-transformed image, guided by knowledge of how emotions are expressed in specific facial regions, FEM aims to create more realistic and accurate facial expressions while preserving the essential features of the original face. This innovative approach goes beyond simple image blending, leveraging domain-specific knowledge to enhance the quality and accuracy of facial expression synthesis results.

This article presents several contributions to the field of facial expression translation as follows:

1. Comprehensive study: We conducted an in-depth investigation of I2I translation, specifically for facial expressions, covering various expressions, including anger, disgust, fear, happiness, sadness, surprise, contempt, and neutrality.

2. Model assessment: We used the Radboud Faces Database (RafD) to evaluate the effectiveness of CycleGAN-based models (CycleGAN, UVCGANv1, and UVCGANv2) for this task.

3. Novel method: FEM-We introduced FEM, a new post-processing technique that enhances the quality and preserves details in facial expression synthesis. FEM specifically addresses the problem of lost facial information, leading to improved results in standard benchmarks. Current methods for translating facial expressions have made significant progress, but they often have difficulty preserving fine facial details and maintaining consistency across different emotions. Our new method, FEM, addresses these limitations by blending input images with translated images to produce the final result. As shown in our comparative results, FEM consistently outperforms existing methods in standard benchmarks and produces visually appealing and expressive results.

It is important to note that our current evaluation is based on the RafD dataset, which contains controlled frontal face images with specific expressions. While FEM shows promising results in this controlled environment, its performance on more diverse and unconstrained datasets still needs to be investigated.

This article is structured as follows: ‘Related Work’ reviews related work in facial expression and image synthesis, and ‘Review of CycleGAN-Based Method’ details the CycleGAN-based model. ‘Method’ details our approach, while ‘Experimental Setup’ outlines the dataset, training process, and evaluation metrics. ‘Result Analysis and Discussion’ presents a thorough analysis of our experimental results, highlighting FEM’s efficacy. ‘Limitations and Future Directions’ discusses Limitations and Future Directions to acknowledge the limitations of our work. Finally, ‘Conclusion’ concludes with a summary of our research and potential future directions.

Related Work

Facial expression synthesis, a critical component of facial attribute editing, is a wide area of research in fields such as computer technology and psychology and has been a challenging topic. Historically, this complex task has often been simplified to a general I2I translation problem, where facial expressions are treated as isolated attributes. Existing methods have focused on manipulating individual facial features, such as adjusting the mouth’s curvature for a smile or opening/closing it (Xia et al., 2021). However, this approach frequently overlooks the subtle interplay of multiple facial features that contribute to authentic expressions, leading to results that may appear artificial or unconvincing.

The Facial Action Coding System (FACS) (Paul & Friesen, 1978) is a widely used, standardized tool for measuring and describing facial expressions. It offers a comprehensive framework for analyzing facial movements based on underlying muscle activity (Action Units), proving invaluable in research fields like psychology, neuroscience, and computer science. FACS is praised for its objectivity, comprehensiveness, scientific validity, and wide range of applications. FACS plays a crucial role in facial expression synthesis by providing a detailed and objective language for defining target expressions and guiding the development of algorithms that can accurately generate and manipulate facial features.

GAN-based models for facial expression synthesis

GANs have emerged as a powerful tool for image synthesis, offering the potential to generate realistic and diverse facial expressions. CycleGAN, a prominent GAN-based model, has been particularly successful in unpaired image-to-image translation tasks, enabling the transformation of images between different domains without requiring paired training data. This capability has been leveraged in various facial expression synthesis studies, demonstrating the effectiveness of CycleGAN in translating expressions while maintaining identity consistency.

Recent studies (LGP-GAN) (Xia et al., 2021) in facial expression synthesis have recognized the limitations of traditional approaches that treated it as a mere image-to-image translation problem, focusing on modifying isolated facial attributes like smiles or mouth shapes. Researchers are now exploring more sophisticated techniques, often grounded in physiology and psychology research, that consider the intricate interplay of multiple facial features. These advancements aim to generate more realistic and expressive faces by capturing the nuanced details that traditional approaches often miss.

One effective approach for facial expression synthesis is LGP-GAN, which introduces an innovative end-to-end model with a two-stage cascaded structure: two local and one global network. In the first stage, the local generators transform the eyes and mouth regions, learning the specific changes associated with different expressions. The second stage utilizes the global network to refine the results by perceiving and supplementing facial information beyond these key areas, resulting in more holistic and realistic facial expressions.

CycleGAN, a cornerstone in computer vision and image processing, is renowned for transforming images across diverse domains through cycle consistency constraints and adversarial training. Its versatility, requiring no explicit supervision or paired image datasets, has been demonstrated through successful applications such as converting photos into paintings, manipulating day and night scenes, or even transforming horses into zebras. This makes it an ideal starting point for exploring facial expression manipulation.

Updated versions of the CycleGAN architecture, UVCGANv1, and UVCGANv2, have been experimented with and evaluated in various image translation tasks. UVCGANv1, incorporating a UNet generator (Ronneberger, Fischer & Brox, 2015) and vision transformers (ViT) (Dosovitskiy et al., 2020), has shown promising results in tasks such as selfie-to-anime, male-to-female, and glasses removal. UVCGANv2, further enhanced with a learnable style token, improves upon these results and expands to additional domains like animal transformations.

However, applying these models to the specific challenge of facial expression translation remains less explored, particularly in scenarios where paired image datasets are unavailable. Our research addresses this underexplored area by comprehensively investigating the use of CycleGAN, UVCGANv1, and UVCGANv2 for translating a wide range of facial expressions using unpaired data, where images of the same person with different expressions are not required. We focus on evaluating the core performance of these models in accurately capturing and generating facial expressions.

While post-processing techniques like GPEN (Tao Yang & Zhang, 2021) could potentially enhance the quality of the translated images, they are not the primary focus of this study, as they might not always guarantee the preservation of identity or accurate expression translation. Moreover, we introduce a novel post-processing step designed specifically to enhance the translated images’ sharpness while prioritizing the preservation of facial identity and the accuracy of the target expression. This contribution is significant as it addresses the limitations of existing GAN-based unpaired I2I translation methods in the context of facial expression synthesis, offering a potential avenue for generating more realistic and accurate results.

Beyond GANs for facial expression synthesis

While GANs have had a significant impact on facial expression synthesis, recent advancements in generative models have opened up new areas for exploration. Diffusion models, known for their gradual denoising process, have attracted attention due to their ability to produce high-quality images with improved stability compared to GANs (Ho, Jain & Abbeel, 2020). Denoising Diffusion Probabilistic Models (DDPMs) (Song, Meng & Ermon, 2020) have demonstrated promising results in various image synthesis tasks, including the creation of realistic faces and manipulation of facial attributes. While their application to targeted facial expression synthesis is still in its early stages, there is potential for these models to generate even more realistic and diverse facial expressions with finer control over subtle nuances.

The Brownian Bridge Diffusion Model (BBDM) (Li et al., 2023) is an important advancement in diffusion models for image-to-image translation. BBDM represents image-to-image translation as a stochastic Brownian bridge process, learning the translation between two domains directly through a bidirectional diffusion process instead of relying on conditional generation. This approach has shown promising results in various image translation tasks, potentially offering a new perspective for addressing challenges in facial expression synthesis, such as preserving identity consistency and handling subtle expression variations.

Moreover, the emergence of text-to-image generation models, such as Stable Diffusion (Rombach et al., 2022) and Midjourney, has expanded the possibilities for creative control in facial expression synthesis. These models enable the generation of images from textual descriptions, providing a new level of flexibility and customization. For example, a user could provide a text prompt like “a surprised face with raised eyebrows and an open mouth” to generate a corresponding image. While challenges exist in accurately capturing subtle nuances and maintaining identity consistency, the potential synergy between text-to-image generation and facial expression synthesis is an exciting area for future research.

However, it is important to note that our current work primarily focuses on exploring the capabilities of GAN-based models for facial expression translation. A comprehensive evaluation of diffusion models and their integration with our proposed FEM method is beyond the scope of this article and will be addressed in future research.

Review of CycleGAN-Based Method

CycleGAN-based models utilize two generator and discriminator pairs to achieve unpaired I2I translation, a valuable technique in this context due to the difficulty of obtaining paired training data. The model connects two generators (GA→B and GB→A) and two discriminators (DA and DB) to translate images between different domains. As shown in Fig. 2, generator GA→B converts images from domain A (e.g., neutral) to domain B (e.g., happy), while GA→B does the reverse. Discriminators DA and DB ensure the quality and realism of the generated expressions by distinguishing between real and translated images.

Figure 2 CycleGAN Framework.

(1) Ldisc=Ex∼pdataxDB→Ax,0+Ez∼pzzDAGB→Az,1

(2) LdiscB=Ex∼pdataxDBx,0+Ez∼pGANzDBGB→Az,1

(3) LGANA=Ex∼pALGANDAGA→Bx,1

(4) Lcyc,A=Ex∼ALregGB→AGA→Bx,x

(5) Lidt,A=Ex∼ALregGB→Ax,x

(6) Lgen,A→B=LGAN,A+λcycLcyc,A+λidtLidt,A

(7) Lgen,B→A=LGAN,B+λcycLcyc,B+λidtLidt,B.

In Eqs. (1) and (2), the discriminator loss (Ldisc) for domains A and B is defined for each input image x. For example, if the input image is a neutral expression (domain A), Ldisc measures how well discriminator DA can distinguish between real neutral expressions from the dataset (𝔼xA) and fake neutral expressions generated by translating happy expressions from domain B (GA→B).

LGAN denotes a classification loss function, such as L2, cross-entropy, or Wasserstein (Arjovsky, Chintala & Bottou, 2017), where translated (fake) images are labeled as 0 and real images as 1. For instance, if the discriminator correctly identifies a generated neutral expression as fake, the loss would be lower than misclassifying it as real.

The generator weights are updated through backpropagation using a combination of three loss components, as shown in Eqs. (6) and (7). These losses are carefully balanced using λ hyperparameters (λcyc and λidt), allowing for fine-tuned control over the relative importance of each component during training. This balancing act ensures that the generator optimizes for multiple objectives simultaneously:

• GAN loss Eq. (3): This encourages the generator to create indistinguishable images from real images, focusing on fooling the discriminator. By minimizing this loss, the generated expressions become increasingly realistic.

• Cycle-consistency loss Eq. (4): This ensures that translating an image from domain A to B and then back again results in an image similar to the original, preserving the subject’s identity. For instance, translating a neutral expression to happy and then back to neutral should ideally result in an image close to the initial neutral expression.

• Identity consistency loss Eq. (5): This encourages the generator to preserve the specific characteristics of the input image, such as facial features, when translating between domains. For example, if the original image has a mole on the cheek, the translated happy expression should ideally retain that mole.

where Lreg refers to any regression loss function, such as L1 or L2, the combination coefficients, λcyc, and λidt, are used to determine the relative weights of the cycle consistency loss and identity loss during training, respectively. By carefully balancing these three components, the generator is guided to produce realistic images that maintain the original content and style while achieving the desired expression transformation.

In this article, we translate facial expressions using the CycleGAN-based approach. Our study uses CycleGAN, UVCGANv1, and UVCGANv2 techniques to translate all facial expressions, including anger, disgust, fear, happiness, sadness, surprise, contempt, and neutrality. This set of expressions can be divided into 28 pairs: angry-to-contempt, angry-to-fear, and so on. Additionally, we improve the output of translation by employing the FEM method as a post-process, which will be explained in detail in the following section.

Method

When using I2I translation models for facial expression synthesis, minor details from the original image, such as skin texture or hairline, can sometimes be lost. Researchers have addressed this issue by proposing a solution in Xia et al. (2021). Their solution involves generating and merging the input and output images, focusing only on the facial regions of interest (ROIs) responsible for the expression.

The Facial Action Coding System (FACS) provides a framework for understanding these ROIs by classifying them and identifying specific action units (AUs) within each section, as shown in Fig. 3. Most expressions primarily involve the eye and mouth areas, except for contempt, which is unique to the mouth area. Understanding which facial regions are most relevant for specific expressions is crucial for developing effective facial expression synthesis algorithms.

Figure 3 FACS action units detail used to identify facial expressions (Paul & Friesen, 1978).

Facial Action Coding System, https://www.cs.cmu.edu/ face/facs.htm.

Table 1 provides a detailed breakdown of the relationship between facial expressions and the specific AUs involved in creating them. Each expression is associated with a combination of AUs, representing the contractions of different facial muscles. For example, a smile (Happiness/Joy) is characterized by the activation of AU 6 (Cheek Raiser) and AU 12 (Lip Corner Puller). Understanding this relationship is crucial for accurately synthesizing facial expressions in digital images.

However, focusing solely on the eye and mouth regions may not capture all facial expressions, particularly those involving forehead movements (AU2 or AU4). To overcome this limitation and the challenge of accurately segmenting facial expression areas, our proposed FEM algorithm preserves the original image’s details while changing its facial expression. This is achieved by generating a new image reflecting the desired expression and selectively merging it with the original image, preserving important details from both.

To address the challenge of preserving facial identity while accurately translating expressions, we propose the FEM algorithm. FEM is a novel post-processing method designed to enhance the visual fidelity of CycleGAN-based outputs by intelligently blending the input image with the generated expression, prioritizing the preservation of crucial facial details. Figure 4 presents a visual overview of the FEM framework, outlining the key steps involved in the process.

Table 1 Relationship between expression and FACS action units.

Expression	Action units (AU)	Description	
Anger	4 + 5 + 7 + 23	Brow Lowerer, Upper Lid Raiser,
Lid Tightener, Lip Tightener	
Contempt	12 + 14 (on one side of the face)	Lip Corner Puller, Dimpler	
Disgust	9 + 15 + 16	Nose Wrinkler, Lip Corner Depressor,	
		Lower Lip Depressor	
Fear	1 + 2 + 4 + 5 + 7 + 20 + 26	Inner Brow Raiser, Outer Brow Raiser,
Brow Lowerer, Upper Lid Raiser,
Lid Tightener, Lip Stretcher, Jaw Drop	
Happiness / Joy	6 + 12	Cheek Raiser, Lip Corner Puller	
Sadness	1 + 4 + 15	Inner Brow Raiser, Brow Lowerer,
Lip Corner Depressor	
Surprise	1 + 2 + 5 + 26	Inner Brow Raiser, Outer Brow Raiser,
Upper Lid Raiser, Jaw Drop	

Figure 4 Overview of FEM framework.

Inspired by face swap (Nirkin, Keller & Hassner, 2019) and face morphing techniques (Venkatesh et al., 2021), the FEM algorithm comprises three main steps: identifying facial expression key points, performing Delaunay triangulation, and finally, morphing the facial expressions. The first step involves detecting and adjusting facial landmarks to capture the unique characteristics of each expression. Next, Delaunay triangulation is employed to create a mesh of triangles over the face, ensuring accurate coverage of the regions of interest. Finally, the facial expression morphing step combines the original and translated images based on the triangulated mesh, resulting in a seamless blend that preserves facial identity while accurately conveying the target expression. The following subsections will provide a detailed explanation of each of these steps.

Facial expression key points

Facial recognition (Kumar, Kaur & Kumar, 2019; Cheng, Hsiao & Lee, 2021) and landmark detection (Wu & Ji, 2019) are computer vision techniques for detecting and analyzing facial images. Landmark detection involves identifying specific points of interest on a face, such as the corners of the eyes, nose, and mouth. We utilize OpenCV (Bradski, 2000), a cross-platform open-source library, to obtain facial boundary boxes and landmark coordinates.

Figure 5 illustrates a facial boundary box and its corresponding landmark key point IDs. The FEM algorithm then modifies these landmark coordinates to create new key points that represent a different facial expression (facial expression key points). Each expression has a unique set of adjustments for the landmark points. All facial landmark key points are used in the transformation process by default.

Figure 5 Facial boundary box and landmark key point ID (the extra key point ID is 69, 70, and 71).

Image source credit: Radboud Faces Database.

If the input or output expression involves forehead movements, we augment the standard facial landmark set with additional key points to ensure accurate representation. Specifically, for AU4 (Brow Lowerer), we add one key point at the top of the face, aligned with the nose tip (ID 34). For AU1 (Inner Brow Raiser), we introduce three extra key points: one at the top center of the face, aligned with the nose tip, and two more positioned above each eyebrow (ID 20 and 25). This augmentation allows a more precise capture of the subtle changes in the forehead region associated with these expressions.

It is important to note that contempt expression is primarily expressed through the lower face. Therefore, for neutral-to-contempt translations, we remove key points below the nose (y-axis lower than key point 30) while maintaining the neutrality of the upper face.

Our approach for generating the final image involves combining key points from both the input and translated images denoted as pnew. We determine a new set of key points (pnew) using Eq. (8), where pori represents the key point from the input image and ptran represents the key point from the translated image. The value of α controls the degree to which the resulting expression resembles the input or translated image. When α is close to 1.0, the result is more similar to the translated image, while an α value close to 0.0 makes the result more similar to the input image. (8) Pnew=1−α×Pori+α×Ptran.

Delaunay triangulation

Delaunay triangulation (Lee & Schachter, 1980) is a computational geometry technique for generating a series of non-overlapping triangles covering a given set of points. It is commonly used in computer graphics, image processing, and other applications. In the context of facial landmarks, Delaunay triangulation can be applied to create a mesh of triangles over an individual’s face.

Specifically, our proposed method uses facial expression key points to create a Delaunay triangulation, ensuring accurate coverage of the regions of interest in each expression pair. This method precisely represents each expression’s unique characteristics and effectively covers both the original (input) and the translated (target expression) images.

Figure 6 Delaunay triangles generated from pori, ptran, and pnew.

α values ranging from 0 to 1 were used, demonstrating the effect of varying morphing degrees on the triangulation. Image source credit: Radboud Faces Database.

For example, in Fig. 6, Delaunay triangles are shown, which are obtained from three sets of key points: pori (original image key points), ptran (translated image key points), and pnew (calculated key points for the morphed image). These triangles are generated with different α values, determining the degree of morphing. A higher α value results in a morphed image that closely resembles the translated expression, while a lower α value results in a morphed image that closely resembles the original image. The specific details of morphing these Delaunay triangles are illustrated in Algorithm 1 .

_______________________ Algorithm 1 Morph_triangulation Algorithm___________________________________________ Require: Imga (Facial image with expression a) Require: Imgb (Translated image in domain b from trained I2I) Require: trianglesa (Delaunay Triangulation from Imga) Require: trianglesb (Delaunay Triangulation from Imgb) Require: triangles (Delaunay Triangulation from combined key points) Require: α (Hyperparameter used to adjust the priority of translated image)   //trianglesa, trianglesb, triangles   warpedImage1 ← CreateEmptyImage(same size as Imga)   warpedImage2 ← CreateEmptyImage(same size as Imgb)   for index ← 0 to size of triangles do       //Get the corresponding triangle from each image        triangle ← triangles[index]        triangle1 ← trianglesa[index]        triangle2 ← trianglesb[index]        //Calculate affine transformation matrices        warpMatrix1to2 ← CalculateAffineTransform(triangle1,triangle2)        warpMatrix2to1 ← CalculateAffineTransform(triangle2,triangle1)        //Warp triangles to a common shape (the average shape)        avgTriangle = CalculateAverageTriangle(triangle1,triangle2)        WarpTriangle(Imga,warpedImage1,triangle1,avgTriangle,warpMatrix1to2)        WarpTriangle(Imgb,warpedImage2,triangle2,avgTriangle,warpMatrix2to1)   end for   morphedImage ← CreateEmptyImage(same size as image1)   for each pixel in morphedImage do       morphedImage[pixel]    =    (1  − α)  × warpedImage1[pixel]  +  α  ×  warpedImage2[pixel]   end for   return morphedImage___________________________________________________________

Facial expression morph

After generating Delaunay triangulations for both the translated and input images, we combine corresponding triangles with the same key points to form the final output structure. This results in a new set of Delaunay triangles made using pnew. While each face has identical key points, the varying shapes and sizes of the original and translated triangles can lead to distortions in the merged result. To address this, we employ affine transformations, which preserve parallelism and ratios of distances, to reshape each merged triangle to match the corresponding Delaunay triangle from pnew, ensuring consistency in appearance.

Once this transformation is applied to all triangles across the face, the resulting image becomes the new expression area. This area is then blended back with the original image using a weighted averaging technique, emphasizing the new expression area more while preserving the overall facial structure and skin tone. Figure 7 illustrates this process, showcasing the FEM framework’s ability to generate realistic expression changes while maintaining facial identity within our selected range of eight basic expressions.

Figure 7 Overview of FEM framework.

Image source credit: Radboud Faces Database.

It is important to note that our experiments focused on frontal face images with these eight basic expressions (anger, disgust, fear, happiness, sadness, surprise, contempt, and neutral), excluding extreme cases that could involve significant distortions of facial features or large head rotations, as well as non-frontal face orientations. These scenarios might require additional adjustments or specialized techniques to ensure seamless blending and prevent artifacts in the final image.

The detailed steps of the FEM algorithm are presented in Algorithm 2 . In the algorithm, detect_landmarks represents Facial Landmark Detection using OpenCV, create_triangulation is used to create Delaunay triangulations from a given set of facial key points, and merge merges two triangles into one, representing a combination of the input and translated images. Finally, Imgoutput represents the final output image that combines the original image with the expression area resulting from morphing two expression areas.

________________________________________________________________________________________ Algorithm 2 FEM Algorithm_______________________________________________________________ Require: Imga (Facial image with expression a) Require: Ma−>b (CycleGAN, UVCGANv1, or UVCGANv2 use to translate expres-    sion a to b) Require: α (Hyperparameter used to adjust the priority of translated image)     Imgb ← Ma−>b(Imga)     Pori ← detect_landmarks((Imga)     Ptran ← detect_landmarks((Imgb)     if a,b = neutral or contempt then       remove keypoint below the nose from Pori and Ptran     else       if a contains AU4 then           Add 1 extra key point to Pori and Ptran         end if       if a contains AU1 then           Add 3 extra key point to Pori and Ptran         end if   end if   Pnew = (1 − α) × Pori + α × Ptran     triangulationori ← create_triangulation(Pori)     triangulationtran ← create_triangulation(Ptran)     triangles =← create_triangulation(Pnew)     ExpressionImage ← Morph_triangulation(triangulationori,triangulationtran,triangles,Imga,Imgb,α)     Imgoutput ← Imga + ExpressionImage   return Imgoutput______________________________________________________________________________________

Experimental Setup

Dataset

Our research involves training a model using the RaFD dataset (Langner et al., 2010), a high-quality collection of images featuring 67 models, including Caucasian males and females, Caucasian children, and Moroccan Dutch males. These images display eight emotional expressions (anger, disgust, fear, happiness, sadness, surprise, contempt, and neutrality) with variations in gaze direction and camera angles (as shown in Fig. 8). The RaFD dataset, created by the Behavioural Science Institute of the Radboud University Nijmegen, is considered ethically sound. The models provided informed consent for their images to be used, and the data is readily available for non-commercial scientific research purposes. This makes the dataset a valuable resource for training robust facial expression recognition models while also supporting transparency and further research in this field.

Figure 8 Example of RaFD images (Langner et al., 2010).

Image source credit: Radboud Faces Database.

While larger datasets like AffectNet (Mollahosseini, Hasani & Mahoor, 2017), containing over 550K images, and RAF-DB (Li, Deng & Du, 2017; Li & Deng, 2019), with 29,672 images, offer potential advantages in terms of sheer volume, we opted for RaFD due to its controlled environment, consistent image quality, and higher resolution. Both AffectNet and RAF-DB include “in-the-wild” images, introducing significant variability in pose, lighting, and occlusions, which can complicate the task of isolating and accurately translating facial expressions. Moreover, despite providing 224 × 224 pixel images (AffectNet) or varying resolutions (RAF-DB), the quality of some images in these datasets can be low. The FER2013 dataset, while popular, suffers from a very low image resolution of 48 × 48 pixels, which can significantly limit the ability to capture and manipulate fine facial details crucial for accurate expression translation. In contrast, RaFD’s controlled setting and high-resolution images facilitate precise analysis and manipulation of facial expressions, aligning well with the objectives of our research.

Before training, we pre-processed the dataset by cropping the images to focus on the face area. We employed an identity-based train-test split to prevent overfitting and ensure the model can generalize to new, unseen individuals. Specifically, we randomly selected seven individuals (approximately 10% of the total) from the dataset and held out all their images to form the testing set. The remaining images from the other 60 individuals constituted the training set. This 90/10 split prioritizes maximizing the training data available to the model, given the relatively small size of the RaFD dataset. This approach guarantees that the model is evaluated on faces it has never encountered during training, providing a more rigorous assessment of its ability to translate expressions on unseen individuals.

• Training set: This subset, typically around 90% of the data, is used to train the model. The model learns to identify patterns and relationships within the data, allowing it to make predictions on new, unseen data.

• Testing set: This smaller subset, usually around 10% of the data, is held out from the training process. It is used to evaluate the model’s performance after training, ensuring that the model can generalize to new faces and expressions it hasn’t seen before.

The RaFD dataset offers a valuable collection of facial expressions for deep learning. However, it is relatively small, which could restrict the model’s ability to adapt to new faces. To address this, we used a specific data augmentation technique: horizontally flipping the images. This method effectively doubled the training data, maintaining the crucial facial expression characteristics and ensuring that the augmented images remained realistic and representative of human faces.

Moreover, we implemented an identity-based train-test split to ensure that the model was assessed on faces it had not encountered during training.

Training detail

Our experiment leveraged three established baseline models with a distinct network architecture: CycleGAN, UVCGANv1, and UVCGANv2. Due to its superior performance in prior research on I2I translation, we adopted the UVCGANv1 approach, avoiding the need for training from scratch. Instead, we pre-trained the generator using self-supervised learning with inpainting (Pathak et al., 2016), a technique where the model learns to restore images by filling in intentionally removed or missing parts. This served as a proxy task for our expression translation goal, as the model effectively learned to reconstruct facial details and textures. This pre-training phase utilized all available expression images, exposing the model to a wide range of facial variations.

Training individual expression pairs were necessary due to the inherent limitations of CycleGAN-based models, which can only transform images between two domains (expressions, in this case) in a single training process. Therefore, we train 28 individual pairs (leading to 56 generators and discriminators), enabling us to focus on refining translations between specific expressions.

Our primary interest was developing and evaluating the FEM algorithm as a post-processing step for CycleGAN-based outputs. Thus, we opted to maintain consistent hyperparameters, adopting the settings from the UVCGANv2 male-to-female translation task (λidt = 0.5, λcyc = 5.0), which demonstrated strong performance in a related facial modification task (male-to-female), rather than performing extensive hyperparameter tuning on the CycleGAN-based model itself.

The training consisted of 500 epochs for each expression pair, utilizing a fixed learning rate of 1e−4 for the first 250 epochs. In the latter half, the learning rate was gradually decreased linearly to facilitate model convergence and refinement. Each model’s training time varied depending on its architecture. CycleGAN took approximately 6 h per expression pair, UVCGANv1 required around 8 h, and UVCGANv2, the most complex model, took approximately 12 h per pair. The specifics of our evaluation metrics and the results obtained will be detailed in the subsequent section, providing a comprehensive assessment of our model’s ability to translate facial expressions accurately and realistically.

Computational environment

All experiments were conducted using the following software and hardware configurations. The deep learning framework utilized was PyTorch 2.1 with CUDA 11.8 and OpenCV 4.9.0. The operating system used was Windows 11 Pro. The hardware configuration included an Intel Core i9-12900F 2.40 GHz, an NVIDIA GeForce RTX 2080 GPU with 12GB VRAM, 16GB DDR4 RAM, and a 2TB NVMe SSD.

Evaluation metric

Metrics for measuring image generation depend on the research purpose. Our goal is to create images that accurately depict the target expression, are realistic, and maintain facial identity. We use different metrics to measure each of these goals. These metrics include translation success rate, Fréchet Inception Distance (FID), and Face Similarity Distance (FSD).

Translation success rate

CycleGAN-based research has demonstrated remarkable success in I2I translation tasks, such as converting images of cats to dogs or transforming male faces into female ones. However, facial expression recognition (FER) (Li & Deng, 2020) presents a unique challenge due to the nuanced details and subtle variations that distinguish different expressions, which may not be fully captured by the CycleGAN model. During our experiments, we observed that the baseline CycleGAN model sometimes struggled to translate certain facial expressions accurately.

To evaluate the effectiveness of I2I translation, we trained a FER model based on EfficientNet B2 (Tan & Le, 2019) using the same train/test split dataset as our I2I models to classify the resulting image and predict the probability of each expression. This FER model achieved 98% accuracy on our test set, demonstrating its strong discriminatory power. Any misclassification was considered a translation failure, and the success/failure ratio was calculated accordingly.

We leverage a FER model based on EfficientNet B2 (Tan & Le, 2019) to evaluate the primary objective of accurate expression translation. This model, trained on the same dataset as our I2I models, classifies the translated image and predicts the probability of each expression. A translation is considered successful only if the predicted probability for the target expression exceeds a predefined threshold of 0.8. This metric directly measures how effectively our method captures and conveys the intended emotion in the generated image.

FID

While the translation success rate focuses on expression accuracy, the FID assesses the overall quality and realism of the generated images. FID leverages a pre-trained Inception-v3 model to extract feature representations from both real and generated images, quantifying their similarity in terms of statistical properties. A lower FID score indicates higher visual fidelity and closer resemblance to real-world images. By incorporating FID into our evaluation, we ensure that the generated facial expressions not only accurately represent the intended emotions but also appear visually convincing and natural. FID is calculated using the following Eq. (9): (9) FID=μ1−μ22+Trσ1+σ2−2σ1σ2

where μ1 and μ2 are the mean vectors of the real and generated image features, and σ1 and σ2 are the covariance matrices of the real and generated image features, respectively. .2 represents the squared L2 norm (Euclidean distance), and Tr denotes the matrix trace.

A lower FID score indicates higher similarity between the two sets of images, suggesting that the generated images are more realistic and diverse. While a score of 0.0 would ideally mean the generated images are indistinguishable from real images, this is often difficult to achieve in practice.

Incorporating FID into our evaluation can ensure that the generated facial expressions accurately represent the intended emotions and appear visually realistic. This is crucial for applications where the quality and believability of generated images are paramount, as FID can help assess the preservation of subtle nuances in facial expression translation.

In addition to FID, we also evaluated our results using the Kernel Inception Distance (KID) metric. However, the trends observed in KID scores consistently aligned with those of FID. To maintain conciseness in the main article, we focus our discussion on FID, but KID results are available upon request.

Face similarity distance

Preserving the original subject’s identity during expression translation is crucial for maintaining realism and avoiding the uncanny valley effect. We utilize the Face Similarity Distance (FSD) metric to evaluate this aspect, which measures the perceptual similarity between the input and output images using a pre-trained FaceNet model (Schroff, Kalenichenko & Philbin, 2015). A lower FSD score signifies greater similarity in facial identity, indicating that the translated image successfully retains the essence of the original person while conveying the target expression. (10) FSD=freal−ftran2.

Equation (10) describes details of the calculation of FSD, where freal represents the feature map of the real input image, and ftran represents the feature map of the translated image. Both feature maps are obtained from FaceNet. Our experiment discusses the differences in FSD scores of each baseline model, both before and after applying our proposed FEM.

Limitations of the metrics

While these metrics offer valuable insights into different aspects of our method’s performance, they also have limitations.

1. Translation success rate: This metric relies on the accuracy of the FER model itself, which might have limitations in recognizing subtle or ambiguous expressions.

2. FID: Primarily focuses on overall image quality and realism but might not fully capture subtle nuances or inconsistencies specific to facial expressions. Moreover, a recent research (Chen & Sub-R-Pa , in press) review shows that FID for I2I is measured in terms of a group of generated images from a specific model. However, defective images can always be clustered when considering each image individually.

3. FSD: While FSD is effective in measuring identity preservation, it might not be sensitive to subtle changes in facial features that could affect the perceived naturalness or expressiveness of the generated image.

In the future, we plan to explore additional metrics or develop new evaluation approaches to address these limitations and provide a more comprehensive assessment of the quality and effectiveness of facial expression translation methods.

Result Analysis and Discussion

Once the model has completed training, it can transform input images into images with the target facial expression. However, to further refine the quality and naturalness of these translated images, we applied our proposed FEM method as a post-processing step.

A key parameter in FEM is α, which determines the balance between maintaining the original facial identity and incorporating the translated expression. We tested α values from 0.6 to 1.0 and provided visual examples in Fig. 9. While all values within this range produced generally acceptable results, we noticed slight differences in the quality and natural appearance of the blended images. When α is set to 1.0, it directly pastes the translated expression onto the original face, which can cause inconsistencies and artifacts, especially when the source and target expressions have significantly different facial shapes. However, lower values like 0.6, while preserving more of the original identity, may not completely capture the subtleties of the target expression. After careful examination, we found that setting α to 0.9 consistently produced the most visually appealing and natural-looking results, striking an optimal balance between preserving the identity and accurately conveying the expression. Therefore, we selected α = 0.9 for all subsequent experiments and analyses presented in this article.

Figure 9 FEM Output in different α setting (happy-to-neutral).

Image source credit: Radboud Faces Database.

With the FEM method finalized (using α = 0.9), we proceeded to assess the CycleGAN-based outputs and the FEM-enhanced results using Translation Success Rate, FID and FSD metrics. Our quantitative analysis focused on the performance of CycleGAN, UVCGANv1, and UVCGANv2, as these models offer a range of CycleGAN-based architectures with different levels of complexity and performance. While other models, such as AttGAN (He et al., 2019), StarGAN (Choi et al., 2018) and GANimation (Pumarola et al., 2018), have been explored in the literature for facial expression translation, their inclusion in our quantitative evaluation was not feasible due to several factors. First, the computationally intensive nature of our experiments, coupled with the limited availability of readily reproducible code for some of these models, posed challenges in conducting a comprehensive comparison. Second, the evaluation settings in the reference articles often differed from ours, such as not including the full forehead in their analysis, making a direct and fair comparison difficult. The following sections provide a detailed analysis of our results, highlighting key insights and findings.

Translation success rate

Tables 2 and 3 shows the success rate of I2I translation for each pair of expressions. Our study indicates that UVCGANv1 and UVCGANv2 can precisely translate all pairs of expressions. However, CycleGAN frequently failed to translate several pairs accurately, including neutral-to-contempt, neutral-to-sad, and others.

Table 2 Translation success rate (CycleGAN/UVCGANv1/UVCGANv2) for neutral, angry, contempt, and disgust.

Values below 0.8 are marked in bold with an asterisk (*).

From \To	Neutral	Angry	Contempt	Disgust	
Neutral	–	0.96/1.0/1.0	*0.19/0.80/0.89	1.0/1.0/1.0	
Angry	0.96/0.85/0.96	–	*0.48/0.89/0.93	1.0/1.0/1.0	
Contempt	*0.78/0.85/0.89	1.0/1.0/1.0	–	0.93/1.0/1.0	
Disgust	1.0/0.96/1.0	1.0/1.0/1.0	*0.74/0.81/0.89	–	
Fearful	0.89/1.0/1.0	0.93/1.0/1.0	*0.26/0.80/0.85	0.81/1.0/1.0	
Happy	1.0/0.96/1.0	1.0/1.0/1.0	*0.78/0.81/0.93	0.96/1.0/1.0	
Sad	*0.52/0.96/0.89	*0.78/1.0/1.0	*0.3/0.81/0.93	0.93/1.0/1.0	
Surprise	0.85/0.93/1.0	0.96/1.0/1.0	*0.63/0.81/1.0	0.93/1.0/1.0	

Table 3 Translation success rate (CycleGAN/UVCGANv1/UVCGANv2) for fearful, happy, sad, and surprise.

Values below 0.8 are marked in bold with an asterisk (*).

From \To	Fearful	Happy	Sad	Surprise	
Neutral	0.93/1.0/1.0	1.0/1.0/1.0	0.7/0.96/1.0	1.0/1.0/0.96	
Angry	0.89/1.0/1.0	0.93/1.0/1.0	1.0/1.0/1.0	1.0/1.0/1.0	
Contempt	0.93/1.0/0.96	0.96/0.93/1.0	*0.74/0.96/1.0	1.0/1.0/1.0	
Disgust	0.89/0.93/1.0	0.93/1.0/1.0	0.78/1.0/1.0	1.0/1.0/1.0	
Fearful	–	0.96/1.0/1.0	0.89/1.0/1.0	1.0/0.96/0.93	
Happy	0.93/1.0/1.0	–	0.93/0.96/1.0	1.0/1.0/1.0	
Sad	*0.74/1.0/1.0	1.0/1.0/1.0	–	0.96/1.0/1.0	
Surprise	0.89/0.96/1.0	0.96/1.0/1.0	1.0/1.0/1.0	-	

We found that the failed images generated by CycleGAN were either blurry or identical to the input image. This is likely due to the model’s inability to differentiate between the input and output expressions, as the generator produced the same output and was trained on it within the same domain due to identity consistency loss. This phenomenon occurs when the model learns to prioritize maintaining the identity of the input image over accurately translating the expression, resulting in minimal changes or no changes at all.

For the CycleGAN generator, we considered 14 combinations as failure translations based on a threshold of 0.8 for the predicted probability of the target expression, which was less than 0.8. However, we did not encounter any issues with UVCGANv1 and UVCGANv2.

FID

In this study, we sought to enhance the realism of facial expressions in images generated by the CycleGAN-based model. To assess and quantify the improvements, we leveraged the FID metric, a well-established measure for evaluating the quality and diversity of generated images. We also evaluate the FID of the images after applying our proposed FEM method as post-processing.

To rigorously evaluate the impact of FEM, we calculated the FID score of the translated image of the testing dataset using the baseline generators (CycleGAN, UVCGANv1, and UVCGANv2), both before and after applying FEM. The resulting FID scores averaged across all input–output expression pairs, are presented in Table 4 for comprehensive analysis. Our results reveal that UVCGANv2, an enhanced version of CycleGAN incorporating the state-of-the-art ViT, consistently outperforms UVCGANv1 and the original CycleGAN regarding FID. This significant improvement underscores the power of ViT architecture in enhancing the model’s ability to generate realistic and accurate facial expressions.

Table 4 The average FID and FSD score for all translated images (the best result was marked as bold).

Model	FID	FSD	
CycleGAN [7] (2017)	74.2525	0.8558	
UVCGAN [8] (2023)	69.8843	0.7478	
UVCGANv2 [9] (2023)	63.8290	0.6941	
CycleGAN + FEM (Our method)	38.8100	0.7630	
UVCGAN + FEM (Our method)	33.4427	0.6806	
UVCGANv2 + FEM (Our method)	31.9271	0.6331	

Moreover, our analysis reveals a consistent pattern: the application of FEM leads to notable improvements in FID scores across all generators. This observation provides compelling evidence of the effectiveness of FEM in enhancing the overall quality of translated facial expression images. Importantly, the benefits of FEM are not confined to a single model but extend across the entire spectrum of evaluated generators.

To delve deeper into the performance of individual models and the specific effects of FEM, we present a comprehensive visual analysis in Fig. 10. This figure showcases the FID values for each input–output expression pair across all generators. The solid bars in the chart represent the FID values obtained without FEM, while the accompanying dotted bars illustrate the FID values after applying FEM.

Figure 10 FID scores for generated images across different input–output expression pairs (all scores on a 0–90 scale).

Solid bars represent FID scores without FEM; dotted bars represent FID scores with FEM applied.

A close examination of Fig. 10 reveals several key insights. First, it confirms the superior performance of UVCGANv2, which consistently outperforms CycleGAN across all expression pairs in terms of FID. Second, there are instances where UVCGANv1 and UVCGANv2 achieve comparable FID scores for specific expression pairs, indicating that the relative performance of these models can be influenced by the specific expression being translated. However, the most striking observation is the consistent and significant improvement in FID across all models after the application of FEM.

The consistent improvement in FID scores across various generators underscores the effectiveness and versatility of FEM. Furthermore, the superior performance of UVCGANv2 highlights the potential of incorporating advanced architectures, such as the ViT, into image translation models.

FSD

Maintaining a person’s identity during facial expression translation is crucial for ensuring the realism and credibility of the generated images. To assess how well our models preserve identity while modifying expressions, we utilize the FSD. This metric quantifies the perceptual similarity between two images by analyzing the distance between their feature representations in a high-dimensional space. A lower FSD score indicates greater similarity, signifying that the translated image retains the essence of the original person while successfully conveying the target expression.

Table 4 comprehensively compares average FSD scores across all translated images, highlighting the models’ ability to preserve identity. We observe that UVCGANv2, the enhanced version of CycleGAN leveraging the ViT, consistently yields lower FSD scores compared to its predecessors. This suggests that UVCGANv2 modifies facial expressions while faithfully maintaining the person’s underlying facial structure and individual characteristics.

Furthermore, the application of our proposed FEM method consistently improves FSD scores across all models. By intelligently blending features from the original and translated images, FEM mitigates unrealistic artifacts that can arise during translation, further enhancing the preservation of identity.

These findings highlight the importance of considering identity preservation alongside expression translation. The results demonstrate that UVCGANv2, particularly when combined with FEM, strikes a remarkable balance between these two competing objectives, generating images that are both expressive and faithful to the original subject.

Qualitative analysis

Visual analysis of the generated images (Fig. 11) revealed that while CycleGAN effectively translated facial expressions, the output images suffered noticeable blurriness, particularly around the mouth. In contrast, UVCGANv1 and UVCGANv2 produced sharper images with higher overall quality, although minor blurring was observed around the skin and chin lines. Notably, our proposed method FEM, a technique where only the facial expression areas are morphed or blended between the translated image and the original image, consistently generated the most visually realistic and convincing results, closely approximating the ground truth images in terms of detail and expression accuracy.

Figure 11 Translation of contempt to neutral expression using different methods (CycleGAN, UVCGANv1, UVCGANv2, and UVCGANv2+FEM).

Image source credit: Radboud Faces Database.

Further examination of expression translations using UVCGANv2 with FEM (Fig. 12) revealed that while our method excelled at generating realistic expressions, certain translations presented challenges. Notably, inconsistencies in preserving personal identity were observed, particularly when transforming sad expressions into happy ones. In contrast, disgust translations were consistently accurate across all samples, capturing the nuanced details and intensity of the emotion. Fear translations proved to be the most difficult, often resulting in distortions around the head and chin due to the significant differences in facial shape between neutral and fearful expressions. Additionally, a minor loss of detail was noted in teeth rendering, likely attributable to limitations in the training data.

Figure 12 Examples of facial expression translation using UVCGANv2+FEM.

Image source credit: Radboud Faces Database.

To assess the performance of our method against established benchmarks, we compared our results (UVCGANv2 + FEM) with those from a recent study, which evaluated AttGAN (He et al., 2019), StarGAN (Choi et al., 2018), GANimation (Pumarola et al., 2018), and LGP-GAN (Xia et al., 2021). Using the same input image and target expression, our method consistently produced images with accurate expressions, sharp details, and minimal blurring compared to other techniques in Fig. 13.

Figure 13 Comparison of facial expression translation results between our method (UVCGANv2+FEM) and methods from [13] (AttGAN, StarGAN, GANimation, LGP-GAN).

Image source credit: Radboud Faces Database.

Despite these strengths, our approach presents challenges. While individual models for each expression pair are not excessively large, the requirement for multiple models leads to a substantial overall storage footprint. Additionally, the need for separate generator models for each expression contributes to a greater cumulative model size, which could be addressed in future work through more efficient training methodologies or model architectures.

In summary, our visual and quantitative analysis demonstrates that our approach, particularly UVCGANv2 with FEM, excels in generating realistic and accurate facial expressions while largely preserving personal identity. However, challenges remain in addressing identity inconsistencies in certain expressions and optimizing computational efficiency. These findings offer valuable insights for future research to advance the state-of-the-art facial expression translation.

Limitations and Future Directions

While our proposed FEM method shows promising results in enhancing the quality and preserving details in facial expression synthesis, we acknowledge certain limitations in the current study.

Dataset

Our experiments primarily utilized the RaFD dataset, which, while comprehensive in terms of expressions and demographics, represents a controlled environment with frontal face images. To ensure the robustness and generalizability of our method, it is crucial to evaluate it on more diverse and unconstrained datasets, including those with varying poses, lighting conditions, and occlusions.

Computational complexity

Our current method implementation requires training individual models for each expression pair, resulting in a substantial overall storage footprint and increased computational complexity during training and inference. Addressing this limitation could involve investigating more efficient training methodologies or model architectures that simultaneously handle multiple expressions.

Identity preservation

While our method is generally successful at preserving facial identity during expression translation, we have observed some inconsistencies, especially when transforming between highly dissimilar expressions (e.g., from sad to happy). In our future research, we aim to enhance our approach to ensure consistent identity preservation across a broader range of expression transformations. Our plans for future work include addressing these limitations in the following ways.

1. Expanding evaluation: We will conduct extensive experiments on additional publicly available facial expression datasets to rigorously assess the generalizability of our method across different facial characteristics and imaging conditions. Moreover, while FID and KID provide valuable insights into image quality and realism, other metrics such as Learned Perceptual Image Patch Similarity (LPIPS), Structural Similarity Index Measure (SSIM) or Recognition Accuracy (RA) (Al-Sumaidaee et al., 2023) could offer complementary perspectives on evaluating facial expression translation. These metrics focus on aspects of image similarity, such as perceptual quality or structural fidelity, which could be relevant for assessing the subtle nuances and identity preservation in generated expressions. Exploring these additional metrics is an area we plan to investigate in future research.

2. Efficiency optimization: We will explore techniques to improve the computational efficiency of our method, potentially through shared model components or more streamlined training procedures.

3. Identity refinement: We will investigate advanced techniques for enhancing identity preservation during expression translation, particularly for challenging cases involving significant facial deformations.

4. Exploring beyond-GANs models: We will investigate the potential of alternative generative models, such as diffusion models, in combination with our FEM approach. This exploration aims to leverage the strengths of different model architectures and potentially achieve even higher levels of realism and accuracy in facial expression synthesis.

By addressing these limitations and exploring new avenues for improvement, we aim to further advance the state of the art in facial expression synthesis and pave the way for its wider adoption in various applications.

Conclusion

In this article, we conducted a comprehensive study on image-to-image translation tasks related to facial expressions, aiming to translate images across a wide range of expressions, including anger, disgust, fear, happiness, sadness, surprise, contempt, and neutrality. Our research focused on evaluating the effectiveness of CycleGAN-based models, namely CycleGAN, UVCGANv1, and UVCGANv2, in achieving accurate and realistic facial expression translations. Our findings demonstrate the superior performance of UVCGANv2, particularly when combined with our proposed FEM method. This novel post-processing technique significantly enhanced the quality of the translated images, as evidenced by improved FID and FSD scores. The enhanced performance of UVCGANv2 can be attributed to its incorporation of the state-of-the-art ViT architecture, which excels at capturing intricate details and nuances in facial expressions.

While our study demonstrates promising results in a controlled setting, future research will address the challenges of translating facial expressions in uncontrolled, real-world environments. We also plan to refine the model further and expand our experiments to include more diverse datasets, ensuring the robustness and generalizability of our approach. By tackling these challenges, we aim to advance the state-of-the-art in facial expression translation and pave the way for innovative applications in various fields, including entertainment, human–computer interaction, and psychology.

Supplemental Information

Supplemental Information 1 Code and instruction.

Supplemental Information 2 Raw data.

Thank you to the data provider.

Additional Information and Declarations

Competing Interests

Author Contributions

Data Availability

The authors declare there are no competing interests.

Chayanon Sub-r-pa conceived and designed the experiments, performed the experiments, analyzed the data, performed the computation work, prepared figures and/or tables, authored or reviewed drafts of the article, and approved the final draft.

Rung-Ching Chen conceived and designed the experiments, analyzed the data, authored or reviewed drafts of the article, and approved the final draft.

Ming-Zhong Fan performed the experiments, prepared figures and/or tables, and approved the final draft.

The following information was supplied regarding data availability:

The RaFD dataset utilized in our experiment is available at https://rafd.socsci.ru.nl/. Researchers are required to request access to the RaFD database individually.

The Facial Action Coding System (FACS) is available at https://www.cs.cmu.edu/ face/facs.htm.

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
