# Peer review of "Facial expression morphing: enhancing visual fidelity and preserving facial details in CycleGAN-based expression synthesis"

_PeerJ Computer Science, doi:10.7717/peerj-cs.2438_

## Round 0.1 · original submission · Major Revisions

Please revise the paper according to the reviewer's comments.

Reviewer 1 ·

Basic reporting

no comment

Experimental design

no comment

Validity of the findings

no comment

Additional comments

1. What are the main contributions of this article? How is it different from existing methods? You need to describe this part of your work in detail in your paper.
2. The Fig. 3 is very unclear, please modify this figure.
3. You should normalize the use of figures and tables.
4. You need to add some contrast to the state-of-the-art methods.

Reviewer 2 ·

Basic reporting

In this paper, the authors undertake a comprehensive study on image-to-image translation tasks pertinent to facial expressions, with the objective of translating images across a spectrum of expressions including anger, disgust, fear, happiness, sadness, surprise, contempt, and neutrality. The research centers on assessing the efficacy of CycleGAN-based models—specifically CycleGAN, UVCGANv1, and UVCGANv2—in achieving accurate and realistic translations of facial expressions.
The study and its subject matter are notably engaging, characterized by well-structured and well-written content. Several measures and analyses have been undertaken, contributing to the overall robustness of the work.

Experimental design

However, the following considerations are crucial before accepting the manuscript:

- Clearly highlight the main contributions of this work in the introduction section using bullet points.
- It is necessary to employ at least a second dataset to guarantee the generalizability of the main findings.
- It is not clear why the authors have split the dataset into 90% for training and 10% for testing sets.
-It is necessary to add a general graphical flowchart to facilitate the comprehension of the proposed framework.

Validity of the findings

- A comparison to some recently published works under the same conditions (i.e., dataset and evaluation protocols) is necessary in the quantitative analysis.

Additional comments

- Figure 3 should be redrawn in a more professional manner.

·

Basic reporting

The article generally meets the standards for basic reporting, but there are a few areas where improvements could be made:

English language and clarity: The paper is mostly well-written, but there are occasional awkward phrasings or minor grammatical issues that could be polished. For example, in the abstract, "Our results, analyzed across CycleGAN, UNet Vision Transformer cycle-consistent GAN versions 1 (UVCGANv1) and 2 (UVCGANv2), demonstrate a significant improvement in sharpness and detail preservation" could be rephrased for better clarity.

Literature references and background: While the paper provides a good overview of related work, it could benefit from a more comprehensive literature review, especially regarding recent advancements in facial expression synthesis beyond GANs. This would help readers better understand the current state of the field and the significance of the proposed method.

Article structure: The overall structure is good, but the "Method" section could be reorganized for better flow. Consider separating the review of CycleGAN-based models from the description of the proposed FEM method for clarity.

Figures and tables: The figures are relevant and well-labeled, but some could be improved in resolution, particularly Figure 1. Additionally, consider adding a table summarizing the key differences between CycleGAN, UVCGANv1, and UVCGANv2 for easier comparison.

Experimental design

Research question and knowledge gap: While the paper clearly presents the problem of preserving facial details in expression synthesis, it could benefit from a more explicit statement of the research question and how it addresses a specific knowledge gap. Consider adding a paragraph in the introduction that clearly outlines the research questions and hypotheses, and explicitly states how this work contributes to the field.

Ethical considerations: The paper doesn't mention any ethical concerns related to using facial images or potential biases in the dataset. Given the sensitive nature of facial data, it would be beneficial to include a brief discussion on the ethical aspects of the research, including privacy considerations and potential biases in the RafD dataset.

Reproducibility: While the methods are generally well-described, some details could be added to enhance reproducibility:

- Provide more specific details on the hardware used for training (eg, GPU models, memory).
- Clarify the exact versions of software libraries used (eg, OpenCV version).
- Consider making the code for the FEM algorithm publicly available, or at least providing pseudocode in the paper.

Experimental setup: The paper could benefit from a more detailed explanation of why certain hyperparameters were chosen (e.g., why α=0.9 for FEM). Additionally, it would be helpful to discuss any limitations of the experimental setup or potential confounding factors.

Baseline comparisons: While the paper compares the proposed method with CycleGAN, UVCGANv1, and UVCGANv2, it could be strengthened by including comparisons with other state-of-the-art methods in facial expression synthesis beyond GAN-based approaches.

Dataset split: More details could be provided on how the dataset was split into training and testing sets. For instance, what was the exact ratio used? How was it ensured that the split was representative of the overall dataset?

Evaluation metrics: While the paper uses several evaluation metrics, it could benefit from a more in-depth discussion of why these specific metrics were chosen and what their limitations might be.

Addressing these points would further strengthen the experimental design and enhance the reproducibility and robustness of the study. Overall, the investigation appears to have been conducted rigorously, but additional details in these areas would improve the paper's scientific contribution.

Validity of the findings

The study presents a novel approach to facial expression synthesis with a clear rationale, addressing the issue of detail loss in generated expressions. However, there are areas for improvement. The paper lacks detailed information about the dataset size and distribution, as well as comprehensive statistical analysis to support the claimed improvements. While the authors compare their method with several baseline models, they don't provide clear justification for these specific choices. The experimental setup could be more detailed to ensure reproducibility. The conclusions are generally well-stated and linked to the research question, but a more thorough discussion of limitations and potential confounding factors would strengthen the paper. To improve, the authors should provide more detailed dataset information, conduct and report statistical tests, include a more comprehensive comparison with other state-of-the-art methods, expand the discussion of limitations, and consider providing access to their code and processed dataset. Addressing these points would significantly bolster the validity of the findings presented in this study.

·

Basic reporting

In general, the manuscrript follows well, howver, there are some typos and insertion of text book general information which affected the quality of presentation is some areas for exampele :
• The proposed method section contains background information such as Facial Expression Key Points method and Delaunay Triangulation and Facial Expression Morph. These should be presented in separate section as they are text book information

• Building upon the previous point, removing these subsection will make the proposed method section too short. Therefore, the authors are required to include the full details of their method to make it replicable.

Experimental design

• Where are the hyperparameters of the proposed method?
• Why the authors have chosen RaFD dataset for testing their method?
• Using 90% of the images for training is not common, this will lead to results bias and overfitting

Validity of the findings

• An analysis of the timing performance and complexity of the proposed method should be included.
• Some benchmark work in the field are missing in related work and/or comparison section such as the work in [A]

[A] Al-Sumaidaee, Saadoon AM, et al. "Spatio-temporal modelling with multi-gradient features and elongated quinary pattern descriptor for dynamic facial expression recognition." Pattern Recognition 142 (2023): 109647.

Additional comments

The authors should address the aforementioned points and provide a point-by-point reply

---

## Round 0.2 · accepted · Accept

According to the comments of reviewers, after comprehensive consideration, it is decided to accept it.

Reviewer 1 ·

Basic reporting

no comment

Experimental design

no comment

Validity of the findings

no comment

Additional comments

no comment

Reviewer 2 ·

Basic reporting

The paper is now in better form. Hence, I would like to accept it.

Experimental design

/

Validity of the findings

/

Additional comments

/

·

Basic reporting

The authors have addressed my previous concerns, no futher comments

Experimental design

The authors have addressed my previous concerns, no futher comments

Validity of the findings

The authors have addressed my previous concerns, no futher comments